# Performance Recovery and Stability Analysis of Disturbance Observer Under Unmodeled Dynamics

**DOI:** 10.3390/s24237850

**Published:** 2024-12-08

**Authors:** Youngjun Joo

**Affiliations:** Department of Electrical Engineering, Institute of Advanced Materials and Systems, Sookmyung Women’s University, Seoul 04310, Republic of Korea; youngjun.joo@sookmyung.ac.kr

**Keywords:** disturbance observer, performance recovery, robust stability, singular perturbation, unmodeled dynamics

## Abstract

Feedback system design is often achieved by neglecting the unmodeled dynamics, such as the actuator and sensor, to reduce design complexity. It is based on an assumption that the unmodeled dynamics are fast enough to be negligible. However, it may cause severe problems for the stability or performance of the overall system, especially, when the controller contains the fast dynamics or uses the high-gain feedback term. A disturbance observer has been widely employed in many industrial applications due to its simple structure and powerful ability to reject disturbances and compensate plant uncertainties. However, since the disturbance observer contains fast dynamics in its structure, the analysis of the effect of the unmodeled dynamics on the disturbance observer-based control is mandatory. This paper reveals the robustness and disturbance rejection performance of the disturbance observer based on the singular perturbation theory and proposes its design guideline for robust stability in the presence of unmodeled dynamics. In addition, this paper presents that the disturbance observer recovers a nominal performance designed for a nominal model of the plant.

## 1. Introduction

In control system design, the existence of disturbances and model uncertainties is unavoidable and can lead to undesired behavior in the closed-loop system. Thus, the disturbance estimation and compensation problem is one of the important issues in the control community, and has been discussed in many applications, such as motor control systems [1,2,3,4,5,6], robot manipulator systems [7,8,9], wheeled mobile robot systems [10], and positioning stage systems [11]. To improve the disturbance rejection performance and robustness, various approaches have been investigated. For instance, an unknown input observer estimates the unknown external signal based on the internal model principle, assuming that the unknown input or disturbance is generated by some exosystem [12,13]. By defining the lumped disturbance as a new state variable, an extended state observer simultaneously estimates the augmented state [14,15]. Further, by incorporating the extended state observer into the control structure, an active disturbance rejection control technique has been presented, and it simplifies the uncertain system with lumped disturbances as a disturbance-free nominal linear system of the form of the integrator chain [16,17]. Instead of canceling the nonlinear term in the nonlinear system, a nonlinear disturbance observer has been proposed to include the known or partially known nonlinear part in the observer design [18,19]. Several survey papers have been presented to figure out the pros and cons of each technique [20,21,22].

Among various disturbance compensation techniques, a disturbance observer has been widely used in the industrial field [23,24,25] because of its versatility. First, it has a simple structure composed of the inverse dynamics of the nominal model of the actual plant and two low-pass filters (known as Q-filter). Secondly, it is convenient to use, since it is an add-on type inner-loop controller. In other words, if it is added to the inner loop, then the existing (pre-designed outer-loop) controller is enabled without taking into account effects from disturbances and plant uncertainties. Thirdly, it effectively compensates the plant uncertainties and rejects external disturbances without knowledge of the disturbance model. Finally, the control designer could easily improve the disturbance rejection performance merely by decreasing the time constant of the Q-filter. Furthermore, by embedding the disturbance model into the disturbance observer structure, the effect of modeled disturbances can be completely removed. A Q-filter design method has been proposed to cope with polynomial-in-time disturbances such as the step, ramp, parabolic signals, and so on [26,27]. It has been extended to deal with sinusoidal-type disturbances with unknown amplitudes and phases [28]. Thus, the disturbance observer can reject not only bounded unmodeled disturbances approximately but also modeled disturbances asymptotically.

Although the characteristics of the disturbance observer is easily understood in the frequency domain [24,29], several papers presented the analysis in the state-space domain for the purpose of obtaining a deeper understanding of the role of each block [30] and extending to the multi-input multi-output nonlinear system [31,32]. They show some interesting points: (1) The external disturbance is almost completely rejected and inner-loop blocks, i.e., the plant with the disturbance observer, behave as the nominal model. (2) the zero dynamics of the plant is replaced by the zero dynamics of the nominal model. This means that the zero dynamics of the plant is nearly unobservable from the output and implies why the zero dynamics should be stable; (3) The disturbance observer recovers the nominal closed-loop system trajectory in the time domain. And (4) the strong ability of the disturbance observer is relevant to an infinite gain property and the structure of the Q-filter is similar to the high-gain observer. Furthermore, an almost necessary and sufficient condition for robust stability of plant with model uncertainties is presented when the cut-off frequency of the Q-filter is high enough. It is also easy to check the stability of the overall system compared with other conditions based on the small-gain theorem [25].

The feedback system design is often achieved by neglecting fast dynamics (e.g., actuator or sensor) to reduce design complexity. It is based on the assumption that the other dynamics, the plant and feedback controller, are much slower than the fast parts. However, when the assumption is not satisfied, i.e., the plant or feedback controller is not slow enough, it causes the degradation of performance, and may lead to instability. Furthermore, the uncertainty and disturbance of the system are important issues for designing a controller. To cope with the uncertainty, various control approaches are considered. Among them, high-gain approaches (e.g., the sliding mode control and the high-gain observer) are very successful in dealing with uncertainty. However, it contains the fast dynamics part in the feedback control loop, and may cause severe problems by the interaction of fast dynamics parts. The work in [33] considers the high-gain full-state feedback, including actuator dynamics. Ref. [34] analyzes the nominal performance and stability recovery for the controller (designed for the system without the actuator dynamics), which is composed of a robust state-feedback controller and high-gain observer. However, it only deals with the single-input single-output nonlinear system without zero dynamics and shows the stability when the actuator dynamics is sufficiently fast. The work in [35] is an extension of [34] to the multi-input multi-output nonlinear system with the sensor dynamics. It also employs the state-feedback control and high-gain observer. The research in [36] shows that the state feedback controller with the high-gain observer is robust under the fast sensor and actuator dynamics, but they do not need to be much faster than the high-gain observer dynamics. Indeed, the disturbance observer contains fast dynamics in its structure as well. Thus, the analysis of the effect of the unmodeled dynamics on the disturbance observer-based control is mandatory.

The contribution of this paper is as follows. First, taking into account the unmodeled dynamics in the closed-loop system, the performance and stability analysis of the disturbance observer-based control system is presented using the singular perturbation theory. Secondly, based on the Lyapunov analysis, robust stability conditions under plant uncertainties are derived. Especially, the explicit bound of the time constant of the Q-filter, which guarantees robust stability of the closed-loop system, is provided. Finally, it is shown that the disturbance observer recovers a nominal performance, which is designed for the nominal model of the plant and the state error between the nominal and the actual closed-loop system asymptotically converges to a set whose size is proportional to the square root of the time constant of the Q-filter. It is important to note that this work is based on the author’s Ph. D. dissertation [37].

The paper is organized as follows. In the following section, motivation and problem formulation are introduced. Section 3 shows an analysis of the disturbance observer when the actuator dynamics are not taken into account in the problem formulation. Stability and performance analysis of DOB with the actuator dynamics are presented in Section 4. Section 5 provides the simulation results to validate the effectiveness of the proposed analysis. The paper concludes with some remarks in Section 6.

**Notation** **1.**
*$0_{k}\in R^{k}$, $0_{j\times k}\in R^{j\times k}$, and $I_{k}\in R^{k\times k}$ denote the zero vector, zero matrix, and identity matrix, respectively. For two column vectors a and b we use $\left[a;b\right]:={\left[\begin{array}{cc} a^{T} & b^{T} \end{array}\right]}^{T}$. $\lambda_{max}\left(E\right)$ and $\lambda_{min}\left(E\right)$ are the maximum and minimum eigenvalue of a matrix E, respectively. The 2-norm of a vector $v\in R^{n}$ is defined by $∥v∥=\sqrt{v^{T}v}$ and the induced 2-norm of a matrix $E\in R^{n\times n}$ of real entries is defined by $∥E∥=\sqrt{\lambda_{max}\left(E^{T}E\right)}$.*


## 2. Problem Formulation and Structure of Disturbance Observer

In this section, we present the disturbance observer-based control structure for an uncertain plant to fulfill the rejection of disturbances and uncertainties and to recover the nominal performance, which is designed for the nominal model of the plant. The configuration of the disturbance observer-based control system is depicted in Figure 1. The block with red dashed line represents the actual plant with the disturbance observer structure. P, $P_{n}$, V, and C denote the actual plant, its nominal model, unmodeled dynamics, and output feedback controller, respectively. $Q_{p}$ and $Q_{q}$ represent the low-pass filters, which are known as the Q-filter. At first, the outer-loop controller is designed for the nominal model to achieve the desired control objectives. Then, the inner-loop controller (i.e., the disturbance observer) is designed so that the actual uncertain plant with the disturbance observer behaves as the disturbance-free nominal model.

Consider the following class of uncertain plants as
(1)$$\begin{array}{cc} \dot{z} & =Sz+Gy,\;y=Cx, \end{array}$$
(2)$$\begin{array}{cc} \dot{x} & =Ax+B\{F_{1}z+F_{2}x+g\left(u+d\right)\}, \end{array}$$
(3)$$\begin{array}{cc} \tau_{v}\dot{v} & =A_{v}v+B_{v}u_{v},\;u=C_{v}v, \end{array}$$
where $x\in R^{\nu }$ and $z\in R^{n-\nu }$ are the states of the plant P, $v\in R^{m}$ is the state of the unmodeled dynamics V, and $u\in R^{1}$, $u_{v}\in R^{1}$, $y\in R^{1}$, and $d\in R^{1}$ are the plant input, control input, plant output, and disturbance, respectively. The matrices *A*, *B*, and *C* are given by
$$\begin{array}{cc} \; & A:=\left[\begin{array}{cc} 0_{\nu -1} & I_{\nu -1}\\ 0 & 0_{\nu -1}^{T} \end{array}\right],\;B:=\left[\begin{array}{c} 0_{\nu -1}\\ 1 \end{array}\right],\;C:=\left[\begin{array}{cc} 1 & 0_{\nu -1}^{T} \end{array}\right]. \end{array}$$
where $\tau_{v}>0$ is a time constant of unmodeled dynamics. The uncertain matrices *S*, *G*, $F_{1}$, $F_{2}$, $A_{v}$, $B_{v}$, and $C_{v}$ are of appropriate dimensions, and *g* is an unknown constant.

**Assumption** **1.**
*The uncertain plant *(1)* and *(2)* satisfy the following assumptions:*
*1.* 
*Uncertain parameters are bounded and their bounds are known a priori. In particular, there exist positive constants ${\bar{F}}_{1}$, ${\bar{F}}_{2}$, $\bar{G}$, $\underline{g}$, and $\bar{g}$ such that $∥F_{1}∥\;\leq \;{\bar{F}}_{1}$, $∥F_{2}∥\;\leq \;{\bar{F}}_{2}$, $∥G∥\;\leq \;\bar{G}$, and $\underline{g}\leq g\leq \bar{g}$.*
*2.* 
*The matrix S is Hurwitz.*
*3.* 
*The disturbance d(t) and its derivative $\dot{d}\left(t\right)$ are bounded with known constants $\phi_{d}$ and $\phi_{dt}$, such that $∥d\left(t\right)∥\leq \phi_{d}$ and $∥\dot{d}\left(t\right)∥\leq \phi_{dt}$.*



Note that, in the absence of the unmodeled dynamics (3), the plant (1) and (2) under consideration is in the normal form whose relative degree is ν (note that a single-input single-output linear time-invariant system whose relative degree ν can always be transformed into the form (1) and (2). For more details, refer to Chapter 13 in [38] and [39]). In addition, the condition 2 in Assumption 1 implies that the plant (1) and (2) is of minimum phase, which is a conventional assumption on the disturbance observer approach.

**Assumption** **2.**
*The unmodeled dynamics in the plant *(3)* are exponentially stable (i.e., the matrix $A_{v}$ is Hurwitz) and $-C_{v}A_{v}^{-1}B_{v}=1$. Furthermore, the time constant $\tau_{v}$ is upper bounded by a positive constant ${\bar{\tau }}_{v}$, which is known a priori.*


The above assumption implies that the subsystem (3) has a unity DC-gain. Even though it is not, a non-unity gain can be integrated into the plant input gain *g*.

Now, we consider a nominal model for the uncertain plant (1), (2), and (3) as follows:(4)$$\begin{array}{c} \begin{array}{cc} {\dot{z}}_{n} & =\bar{S}z_{n}+\bar{G}y_{n},\;y_{n}=Cx_{n},\\ {\dot{x}}_{n} & =Ax_{n}+B\left\{{\bar{F}}_{1}z_{n}+{\bar{F}}_{2}x_{n}+g_{n}u_{r}\right\}, \end{array} \end{array}$$
where $x_{n}\in R^{\nu }$ and $z_{n}\in R^{\bar{n}-\nu }$ are the state, and $u_{r}\in R^{1}$ and $y_{n}\in R^{1}$ are the control input and output of the nominal model, respectively. Notice that the order of the nominal zero dynamics $z_{n}$ may not be equal to that of the zero dynamics (Equation 1), i.e., $\bar{n}$ may not be equal to *n*. $\bar{S}$, $\bar{G}$, ${\bar{F}}_{1}$, ${\bar{F}}_{2}$, and $g_{n}$ are the nominal values of *S*, *G*, $F_{1}$, $F_{2}$, and *g*, respectively. For the nominal model (Equation 4), we design an output feedback controller (i.e., outer-loop controller) as
(5)$$\begin{array}{c} \begin{array}{cc} \dot{\eta } & =A_{\eta }\eta +B_{\eta }y_{r}-E_{\eta }y_{n},\\ u_{r} & =C_{\eta }\eta +D_{\eta }y_{r}-H_{\eta }y_{n} \end{array} \end{array}$$
where $\eta \in R^{h}$ is the state of output feedback controller and $y_{r}$ is the reference input. The matrices $A_{\eta }$, $B_{\eta }$, $C_{\eta }$, $D_{\eta }$, $E_{\eta }$, and $H_{\eta }$ are of appropriate dimensions. It is assumed that $y_{r}\left(t\right)$ and ${\dot{y}}_{r}\left(t\right)$ are bounded with known bounds $\phi_{r}$ and $\phi_{rt}$, such that $∥y_{r}\left(t\right)∥\leq \phi_{r}$ and $∥{\dot{y}}_{r}\left(t\right)∥\leq \phi_{rt}$, respectively.

**Assumption** **3.**
*The nominal closed-loop system *(Equation 4)* and *(Equation 5)* is exponentially stable. It implies that it is input-to-state stable with respect to the reference input $y_{r}$.*


We will show that the plant (1), (2), and (3) with the disturbance observer behaves as the disturbance-free nominal model (4) in the presence of disturbances and plant uncertainties. Therefore, the outer-loop controller (5) has to be designed to stabilize the nominal model (4) and the specific design of (5) is determined by the control objectives (e.g., tracking or regulation).

The disturbance observer as an inner-loop controller is represented by
(6)$$\begin{array}{cc} \dot{\bar{z}} & =\bar{S}\bar{z}+\bar{G}\bar{w},\;w=\frac{1}{g_{n}}\left(-{\bar{F}}_{1}\bar{z}-{\bar{F}}_{2}{\bar{w}}^{†}+{\bar{w}}^{\nu }\right), \end{array}$$
(7)$$\begin{array}{cc} \dot{q} & =A_{q}\left(\tau_{q}\right)q+B_{q}y,\;\bar{w}=C_{q}\left(\tau_{q}\right)q, \end{array}$$
(8)$$\begin{array}{cc} \dot{p} & =A_{q}\left(\tau_{q}\right)p+B_{q}u_{v},\;\hat{u}=C_{q}\left(\tau_{q}\right)p, \end{array}$$
(9)$$\begin{array}{cc} u_{v} & =u_{r}+\hat{u}-w \end{array}$$
where $\bar{z}\in R^{\bar{n}-\nu }$, $q\in R^{l}$, and $p\in R^{l}$ are the state of $P_{n}^{-1}$, $Q_{q}$, and $Q_{p}$, respectively, ${\bar{w}}^{†}={\left[\begin{array}{cccc} \bar{w} & \dot{\bar{w}} & \cdots & {\bar{w}}^{\nu -1} \end{array}\right]}^{T}$, and ${\bar{w}}^{i}$ is the *i*-th derivative of $\bar{w}$. The matrices $A_{q}\left(\tau_{q}\right)$, $B_{q}$, and $C_{q}\left(\tau_{q}\right)$ are
$$\begin{array}{cc} A_{q}\left(\tau \right) & :=\left[\begin{array}{cccc} 0 & 1 & \cdots & 0\\ 0 & 0 & \cdots & 0\\ \vdots & \vdots & \ddots & \vdots \\ 0 & 0 & \cdots & 1\\ -\frac{a_{0}}{\tau_{q}^{l}} & -\frac{a_{1}}{\tau_{q}^{l-1}} & \cdots & -\frac{a_{l-1}}{\tau_{q}} \end{array}\right],\;B_{q}:=\left[\begin{array}{c} 0\\ 0\\ \vdots \\ 0\\ 1 \end{array}\right],\\ C_{q}\left(\tau \right) & :=\left[\begin{array}{ccccccc} \frac{c_{0}}{\tau_{q}^{l}} & \frac{c_{1}}{\tau_{q}^{l-1}} & \cdots & \frac{c_{k}}{\tau_{q}^{l-k}} & 0 & \cdots & 0 \end{array}\right]. \end{array}$$
where l−k≥ν, $c_{0}=a_{0}$, and all $a_{i}$’s are chosen such that the polynomial $s^{l}+a_{l-1}s^{l-1}+\cdots +a_{1}s+a_{0}$ is Hurwitz. The detailed design procedure for coefficients $a_{i}$, $c_{i}$, and $\tau_{q}$ will be discussed later.

It is important to note that the disturbance observer in (6), (7), (8), and (9) is a state-space realization of the conventional disturbance observer, which is usually employed in the frequency domain [23,24,25,29]. The dynamics (6) has the same structure as an inverse dynamics of (4) and the dynamics (7) and (8) are the controllable canonical form realizations of a stable low-pass filter, which is known as Q-filter. In addition, since l−k≥ν, the signal ${\bar{w}}^{\nu }$ and ${\bar{w}}^{†}$ can be implemented from the state of (7) and the output *y*. In fact, an exact inverse dynamics of (4) is calculated as
(10)$$\begin{array}{c} \begin{array}{cc} {\dot{z}}_{n} & =\bar{S}z_{n}+\bar{G}y_{n},\\ u_{r} & =\frac{1}{g_{n}}\left(-{\bar{F}}_{1}z_{n}-{\bar{F}}_{2}y_{n}^{†}+y_{n}^{\nu }\right) \end{array} \end{array}$$
where $y_{n}^{†}={\left[\begin{array}{cccc} y_{n} & {\dot{y}}_{n} & \cdots & y_{n}^{\nu -1} \end{array}\right]}^{T}$ and $y_{n}^{i}$ is the *i*-th derivative of the output $y_{n}$. However, it is not realizable alone since the dynamics (10) contains the derivative of $y_{n}$. Therefore, the dynamics (6) and (7) whose input *y* and output *w* is implemented together to prevent the derivative of the output. Let us exchange the dynamics (6) with (7) as follows: (11)$$\begin{array}{cc} \dot{\bar{z}} & =\bar{S}\bar{z}+\bar{G}y,\;\bar{w}=\frac{1}{g_{n}}\left(-{\bar{F}}_{1}\bar{z}-{\bar{F}}_{2}y^{†}+y^{\nu }\right), \end{array}$$(12)$$\begin{array}{cc} \dot{q} & =A_{q}\left(\tau_{q}\right)q+B_{q}\bar{w},\;w=C_{q}\left(\tau_{q}\right)q, \end{array}$$(13)$$\begin{array}{cc} \dot{p} & =A_{q}\left(\tau_{q}\right)p+B_{q}u_{v},\;\hat{u}=C_{q}\left(\tau_{q}\right)p, \end{array}$$(14)$$\begin{array}{cc} u_{v} & =u_{r}+\hat{u}-w \end{array}$$
where $y^{†}=\left[\begin{array}{cccc} y & \dot{y} & \cdots & y^{\nu -1} \end{array}\right]$ and $y^{i}$ is the *i*-th derivative of the output *y*. By virtue of the linearity, the input–output behavior between *y* and *w* of (6) and (7) is the same as that of (11) and (12). Throughout this paper, for simplicity, the dynamics (11), (12), (13), and (14) is used instead of (6), (7), (8), and (9) although the time response of *q* in (11), (12), (13), and (14) is different from that of (6), (7), (8), and (9).

In order to obtain a singular perturbation form, we change coordinates for states *q* and *p* as follows:(15)$$\begin{array}{c} \begin{array}{cc} \; & \xi_{i}:=\tau_{q}^{i-(l+1)}q_{i},\;\zeta_{i}:=\tau_{q}^{i-(l+1)}p_{i}. \end{array} \end{array}$$

With (Equation 15), the dynamics of ξ, ζ, and *v* are represented as
(16)$$\begin{array}{c} \begin{array}{cc} \left[\begin{array}{c} \tau_{q}\dot{\xi }\\ \tau_{q}\dot{\zeta }\\ \tau_{v}\dot{v} \end{array}\right] & =A_{u}\left[\begin{array}{c} \xi \\ \zeta \\ v \end{array}\right]+\left[\begin{array}{c} \frac{1}{g_{n}}B_{\xi }\left\{\tilde{F}\left(z,\bar{z},x\right)+gd\right\}\\ B_{\xi }u_{r}\\ B_{v}u_{r} \end{array}\right], \end{array} \end{array}$$
where
$$A_{u}:=\left[\begin{array}{ccc} A_{\xi } & O_{l\times l} & \frac{g}{g_{n}}B_{\xi }C_{v}\\ -B_{\xi }C_{\xi } & A_{\xi }+B_{\xi }C_{\xi } & O_{l\times m}\\ -B_{v}C_{\xi } & B_{v}C_{\xi } & A_{v} \end{array}\right],$$
where $\tilde{F}\left(z,\bar{z},x\right):=-{\bar{F}}_{1}\bar{z}-{\bar{F}}_{2}x+F_{1}z+F_{2}x$, and $A_{\xi }$, $B_{\xi }$, and $C_{\xi }$ imply $A_{q}$, $B_{q}$, and $C_{q}$ when $\tau_{q}=1$, respectively.

Then, from Equations (1), (2), (3), (5), (11) and (16), the overall closed-loop system can be written as follows ( note that, when the output-lop controller (5) is considered in the overall closed-loop system, $y_{n}$ should be replaced by *y*, which is evident and will be applied without mention throughout the paper. Furthermore, $u_{r}$, the function of *y*, η, and $y_{r}$, will be used for simplification of equations):and
(17)$$\begin{array}{c} \begin{array}{cc} \dot{\eta } & =A_{\eta }\eta +B_{\eta }y_{r}-E_{\eta }y,\;u_{r}=C_{\eta }\eta +D_{\eta }y_{r}-H_{\eta }y,\\ \dot{x} & =Ax+B\{F_{1}z+F_{2}x+g\left(C_{v}v+d\right)\},\\ \dot{z} & =Sz+Gy,\\ \dot{\bar{z}} & =\bar{S}\bar{z}+\bar{G}y,\;y=Cx, \end{array} \end{array}$$
and
(18)$$\begin{array}{c} \begin{array}{c} \left[\begin{array}{c} \tau_{q}\dot{\xi }\\ \tau_{q}\dot{\zeta }\\ \tau_{v}\dot{v} \end{array}\right]=A_{u}\left[\begin{array}{c} \xi \\ \zeta \\ v \end{array}\right]+\left[\begin{array}{c} \frac{1}{g_{n}}B_{\xi }\left\{\tilde{F}\left(z,\bar{z},x\right)+gd\right\}\\ B_{\xi }u_{r}\\ B_{v}u_{r} \end{array}\right]. \end{array} \end{array}$$

From the overall closed-loop system (Equation 17) and (Equation 18), it is observed that, for relatively small $\tau_{q}$ and $\tau_{v}$, the system is in the multi-parameter or the multi-time-scale singular perturbation form (if time constants $\tau_{q}$ and $\tau_{v}$ are in same order, then the system is in the multi-parameter singular perturbation form [40,41]. Otherwise, it is in the multi-time-scale singular perturbation form [42]).

## 3. Analysis of Disturbance Observer Based on Singular Perturbation Approach

In this section, we will discuss the disturbance rejection performance of the disturbance observer based control. It is observed from (Equation 17) and (Equation 18) that the variables *x*, *z*, $\bar{z}$, η, $y_{r}$, and *d* are considered as slow variables, while the state ξ, ζ, and *v* are regarded as fast variables. If the fast dynamics has an isolated equilibrium for each (frozen) slow variables and the equilibrium (depending on slow variables) is exponentially stable (i.e., the matrix $A_{u}$ is Hurwitz), then the overall closed-loop system behaves as the reduced system. Namely, the overall closed-loop system is restricted to the slow manifold, with sufficiently small $\tau_{q}$ and $\tau_{v}$ under the assumption that the slow variables are bounded and varying slowly. In order to reveal that the disturbance observer recovers the steady-state performance of the nominal closed-loop system (Equation 4) and (Equation 5), we obtain the quasi-steady-state subsystem from (Equation 17) and (Equation 18) for the extreme case $\tau_{q}=\tau_{v}=0$.

The equilibrium of (Equation 17) and (Equation 18) for each frozen slow variable is simply computed as
(19)$$\begin{array}{cc} \left[\begin{array}{c} \xi^{*}\\ \zeta^{*}\\ v^{*} \end{array}\right] & =-A_{u}^{-1}\left[\begin{array}{c} \frac{1}{g_{n}}B_{\xi }\left\{\tilde{F}\left(z,\bar{z},x\right)+gd\right\}\\ B_{\xi }u_{r}\\ B_{v}u_{r} \end{array}\right]. \end{array}$$

With the help of the matrix inversion lemma, each equilibrium is rewritten as
(20)$$\begin{array}{cc} \xi^{*} & =-\frac{g_{n}+g}{g_{n}}{\left(A_{\xi }-\frac{g}{g_{n}}B_{\xi }C_{\xi }\right)}^{-1}B_{\xi }u_{r}, \end{array}$$
(21)$$\begin{array}{cc} \zeta^{*} & =\frac{1}{g_{n}+g}{\left(A_{\xi }+\frac{g_{n}}{g_{n}+g}B_{\xi }C_{\xi }\right)}^{-1}B_{\xi }\left\{\tilde{F}\left(z,\bar{z},x\right)+gd-g_{n}u_{r}\right\}, \end{array}$$
(22)$$\begin{array}{cc} v^{*} & =\frac{1}{g}A_{v}^{-1}B_{v}\left\{\tilde{F}\left(z,\bar{z},x\right)+gd-g_{n}u_{r}\right\}. \end{array}$$

With these equilibria, we derive the quasi-steady-state system (i.e., slow dynamics on the slow manifold when $\tau_{q}=\tau_{v}=0$) as follows:(23)$$\begin{array}{c} \begin{array}{cc} \dot{\eta } & =A_{\eta }\eta +B_{\eta }y_{r}-E_{\eta }y,\;u_{r}=C_{\eta }\eta +D_{\eta }y_{r}-H_{\eta }y,\\ \dot{x} & =Ax+B\{{\bar{F}}_{1}\bar{z}+{\bar{F}}_{2}x+g_{n}u_{r}\},\\ \dot{\bar{z}} & =\bar{S}\bar{z}+\bar{G}Cx,\\ \dot{z} & =Sz+GCx,\;y=Cx. \end{array} \end{array}$$

The quasi-steady-state system (Equation 23) play the key role to explain the nominal performance recovery of the disturbance observer based control. From (Equation 23), we find out several interesting points. First, the input disturbance *d* is completely rejected from the control input. Second, in the viewpoint of the outer-loop controller (Equation 5), the plant to be controlled is approximated as the nominal model (Equation 4) that is completely known to the controller. Finally, the zero dynamics of the plant (i.e., *z*-dynamics of (Equation 1)) is disconnected from the output *y*, i.e., becomes unobservable from the output. In fact, it is replaced by the zero dynamics of the nominal model (i.e., $\bar{z}$-dynamics of (Equation 4)). For more details on (Equation 23), please see [30].

Now, we analyze robust stability for the overall closed-loop system (Equation 17) and (Equation 18) based on the singular perturbation approach with respect to the ratio between $\tau_{q}$ and $\tau_{v}$. When $\tau_{v}\ll \tau_{q}$ (i.e., the unmodeled dynamics is much faster than *p* and *q*-dynamics), (Equation 17) and (Equation 18) can be considered as the three-time scale singular perturbation form.

**Theorem** **1.**
*Under Assumptions 1–3, there exists a positive constant ${\bar{\tau }}_{q}$ such that, for all $0<\tau_{v}\ll \tau_{q}<{\bar{\tau }}_{q}$, the overall closed-loop system *(Equation 17)* and *(Equation 18)* is robustly exponentially stable if the matrix $A_{f}$*

(24)
$$A_{f}:=\left[\begin{array}{cc} A_{\xi }-\frac{g}{g_{n}}B_{\xi }C_{\xi } & \frac{g}{g_{n}}B_{\xi }C_{\xi }\\ -B_{\xi }C_{\xi } & A_{\xi }+B_{\xi }C_{\xi } \end{array}\right]$$

*is Hurwitz for all uncertain g.*


**Proof** **of** **Theorem** **1.**Since $\tau_{v}\ll \tau_{q}$, we consider *v*-dynamics in (Equation 17) and (Equation 18) as fast dynamics, while the other dynamics are slow dynamics. From the singular perturbation theory, if both the quasi-steady-state and the boundary-layer subsystem are exponentially stable, then the overall closed-loop system is exponentially stable. By Assumption 2, it follows that the boundary-layer subsystem (*v*-dynamics) is exponentially stable.In the next step, we will show that the quasi-steady-state subsystem is exponentially stable. Since $C_{v}A_{v}^{-1}B_{v}=-1$, the quasi-steady-state system is easily calculated as follows:
(25)$$\begin{array}{c} \begin{array}{cc} \dot{\eta } & =A_{c}\eta +B_{c}r-E_{c}y,\;u_{r}=C_{c}\eta +D_{c}r-H_{c}y,\\ \dot{x} & =Ax+B\{F_{1}z+F_{2}x+gC_{\xi }\left(\zeta -\xi \right)+gu_{r}+gd\},\\ \dot{z} & =Sz+Gy,\\ \dot{\bar{z}} & =\bar{S}\bar{z}+\bar{G}y,\;y=Cx, \end{array} \end{array}$$
and
(26)$$\begin{array}{c} \begin{array}{cc} \tau_{q}\left[\begin{array}{c} \dot{\xi }\\ \dot{\zeta } \end{array}\right] & =A_{f}\left[\begin{array}{c} \xi \\ \zeta \end{array}\right]+\left[\begin{array}{c} \frac{1}{g_{n}}B_{\xi }\left\{\tilde{F}\left(z,\bar{z},x\right)+gu_{r}+gd\right\}\\ B_{\xi }u_{r} \end{array}\right], \end{array} \end{array}$$Now, it can be observed that (Equation 25) and (Equation 26) is the two-time scale singular perturbation form. The dynamics (Equation 25) and (Equation 26) are considered as slow and fast dynamics, respectively. After a simple calculation with $\tau_{q}=0$, it is easy to see that (Equation 23) is its quasi-steady-state subsystem. From Assumption 1 and 3, it follows that (Equation 23) is exponentially stable. The proof is completed since the matrix $A_{f}$ is Hurwitz. □

It is emphasized that the matrix $A_{f}$ plays a key role to determine the stability of the overall closed-loop system (Equation 17) and (Equation 18). If it is satisfied, then (Equation 17) and (Equation 18) is robustly stable for the sufficiently small $\tau_{q}$. The next lemma shows the design procedure for $a_{i}$ and $c_{i}$ to make $A_{f}$ Hurwitz.

**Lemma** **1.**
*The matrix $A_{f}$ is Hurwitz if and only if the following two polynomials are Hurwitz:*

$$\begin{array}{c} \begin{array}{cc} p_{a}\left(s\right) & =s^{l}+a_{l-1}s^{l-1}+\cdots +a_{1}s+a_{0},\\ p_{b}\left(s\right) & =s^{l}+a_{l-1}s^{l-1}+\cdots +a_{k+1}s^{k+1}+\left(a_{k}-\frac{g-g_{n}}{g_{n}}c_{k}\right)s^{k}+\cdots +\left(a_{0}+\frac{g-g_{n}}{g_{n}}c_{0}\right). \end{array} \end{array}$$



The proof of Lemma 1 is presented in [30]. The detailed procedure so as to make the matrix $A_{f}$ Hurwitz (i.e., selecting $a_{i}$ and $c_{i}$) was discussed in [28].

**Remark** **1.**
*When the dynamics of Q-filter is much faster than the unmodeled dynamics v (i.e., $\tau_{q}\ll \tau_{v}$), the stability of the overall closed-loop system is not guaranteed. Since $\tau_{q}\ll \tau_{v}$, the dynamics of Q-filter are considered as fast dynamics. Then, from *(Equation 17)* and *(Equation 18)*, the system matrix of ξ and ζ becomes*

$$\begin{array}{c} \left[\begin{array}{cc} A_{\xi } & O_{l\times l}\\ -B_{\xi }C_{\xi } & A_{\xi }+B_{\xi }C_{\xi } \end{array}\right] \end{array}$$

*and always has one eigenvalue at the origin. Therefore, the singular perturbation theory cannot be employed since the boundary-layer system is not exponentially stable. In fact, if the relative degree of v-dynamics is greater than one, then robust stabilization is impossible when the time constant $\tau_{q}$ is much smaller than $\tau_{v}$ [43].*


## 4. Nominal Performance Recovery by Disturbance Observer Under Unmodeled Dynamics

This section presents an explicit bound of $\tau_{q}$ to guarantee robust stability of the disturbance observer approach. Furthermore, it is shown that the disturbance and uncertainty compensation performance of the disturbance observer is improved as $\tau_{q}$ becomes smaller. In other word, the closed-loop system with the disturbance observer recovers the nominal performance with sufficiently small $\tau_{q}$.

The closed-loop system (Equation 17) and (Equation 18) can be compactly rewritten as
(27)$$\begin{array}{c} \begin{array}{cc} \dot{X} & ={\bar{A}}_{s}X+A_{xv}Z_{2}+{\bar{B}}_{x}V,\\ \tau_{q}\dot{Z_{1}} & =\bar{A_{f}}Z_{1}+{\bar{A}}_{qx}X+A_{qv}Z_{2}+{\bar{B}}_{q}V,\\ \tau_{v}{\dot{Z}}_{2} & =A_{v}Z_{2}+A_{vx}X+A_{vq}Z_{1}+B_{v}V \end{array} \end{array}$$
where $X=[\eta ;x;z;\bar{z}]$, $Z_{1}:=\left[\xi ;\zeta \right]$, $Z_{2}:=v$, V:=[r;d], and the matrices ${\bar{A}}_{s}$, $A_{xv}$, ${\bar{A}}_{f}$, ${\bar{A}}_{qx}$, $A_{qv}$, $A_{vx}$, $A_{vq}$, ${\bar{B}}_{x}$, ${\bar{B}}_{q}$, and $B_{v}$ are
$$\begin{array}{cc} \; & {\bar{A}}_{s}:=\left[\begin{array}{cccc} A_{\eta } & -E_{\eta }C & 0_{h\times n-\nu } & 0_{h\times \bar{n}-\nu }\\ 0_{\nu \times h} & A+BF_{2} & BF_{1} & 0_{\nu \times \bar{n}-\nu }\\ 0_{n-\nu \times h} & GC & S & 0_{n-\nu \times \bar{n}-\nu }\\ 0_{\bar{n}-\nu \times h} & \bar{G}C & 0_{\bar{n}-\nu \times n-\nu } & \bar{S} \end{array}\right],\;A_{xv}:=\left[\begin{array}{c} 0_{h\times m}\\ gBC_{v}\\ 0_{n-\nu \times m}\\ 0_{\bar{n}-\nu \times m} \end{array}\right],\\ \; & {\bar{A}}_{f}:=\left[\begin{array}{cc} A_{\xi } & 0_{l\times l}\\ -B_{\xi }C_{\xi } & A_{\xi }+B_{\xi }C_{\xi } \end{array}\right],\;A_{vx}:=\left[\begin{array}{cccc} B_{v}C_{c} & -B_{v}H_{c}C & 0_{m\times n-\nu } & 0_{m\times \bar{n}-\nu } \end{array}\right],\\ \; & A_{qv}:=\left[\begin{array}{c} \frac{g}{g_{n}}B_{\xi }C_{v}\\ 0_{l\times m} \end{array}\right],\;{\bar{A}}_{qx}:=\left[\begin{array}{cccc} 0_{l\times h} & \frac{1}{g_{n}}B_{\xi }\left[-{\bar{F}}_{2}+F_{2}\right] & \frac{1}{g_{n}}B_{\xi }F_{1} & -\frac{1}{g_{n}}B_{\xi }{\bar{F}}_{1}\\ B_{\xi }C_{\eta } & -B_{\xi }H_{\eta }C & 0_{l\times n-\nu } & 0_{l\times \bar{n}-\nu } \end{array}\right],\\ \; & A_{vq}:=\left[\begin{array}{cc} -B_{v}C_{\xi } & B_{v}C_{\xi } \end{array}\right],\;{\bar{B}}_{x}:=\left[\begin{array}{cc} B_{\eta } & 0_{h}\\ 0_{\nu } & gB\\ 0_{n-\nu } & 0_{n-\nu }\\ 0_{\bar{n}-\nu } & 0_{\bar{n}-\nu } \end{array}\right],\;{\bar{B}}_{q}:=\left[\begin{array}{cc} 0_{l} & \frac{g}{g_{n}}B_{\xi }\\ B_{\xi }D_{c} & 0_{l} \end{array}\right],\\ \; & B_{v}:=\left[\begin{array}{cc} B_{v}D_{\eta } & 0_{m} \end{array}\right]. \end{array}$$

Let $h_{1}\left(X,V\right):=-A_{f}^{-1}\left(A_{qx}X+B_{q}V\right)$ and $h_{2}\left(X,Z_{1},V\right):=-A_{v}^{-1}\left(A_{vx}X+A_{vq}Z_{1}+B_{v}V\right)$ where
$$\begin{array}{cc} \; & A_{qx}:=\left[\begin{array}{cccc} \frac{g}{g_{n}}B_{\xi }C_{c} & \frac{1}{g_{n}}B_{\xi }\left[-{\bar{F}}_{2}+F_{2}-gH_{c}C\right] & \frac{1}{g_{n}}B_{\xi }F_{1} & -\frac{1}{g_{n}}B_{\xi }{\bar{F}}_{1}\\ B_{\xi }C_{c} & -B_{\xi }H_{c}C & 0_{l\times n-\nu } & 0_{l\times \bar{n}-\nu } \end{array}\right],\\ \; & B_{q}:=\left[\begin{array}{cc} \frac{g}{g_{n}}B_{\xi }D_{c} & \frac{g}{g_{n}}B_{\xi }\\ B_{\xi }D_{c} & 0_{l\times 1} \end{array}\right]. \end{array}$$

In fact, $h_{1}\left(X,V\right)=\left[\xi^{*};\zeta^{*}\right]$ and $h_{2}\left(X,Z_{1},V\right)=v^{*}$ when $Z_{1}=h_{1}\left(X,V\right)$, which are the equilibrium in (Equation 20). With $Y_{1}:=Z_{1}-h_{1}\left(X,V\right)$ and $Y_{2}:=Z_{2}-h_{2}\left(X,Z_{1},V\right)$, we have
(28)$$\begin{array}{c} \begin{array}{cc} \dot{X} & =F_{s}X+A_{xq}Y_{1}+A_{xv}Y_{2}+\left(B_{x}-A_{xq}A_{f}^{-1}B_{q}\right)V\\ \dot{Y_{1}} & =F_{q}X+\left(\frac{1}{\tau }A_{f}+A_{f}^{-1}A_{qx}A_{xq}\right)Y_{1}+\left(\frac{1}{\tau }A_{qv}+A_{f}^{-1}A_{qx}A_{xv}\right)Y_{2}\\ \; & \;+A_{f}^{-1}A_{qx}\left(B_{x}-A_{xq}A_{f}^{-1}B_{q}\right)V+A_{f}^{-1}B_{q}\dot{V}\\ \dot{Y_{2}} & =F_{v}X+\left(\frac{1}{\tau }A_{v}^{-1}A_{vq}A_{f}+A_{v}^{-1}A_{vx}A_{xq}\right)Y_{1}\\ \; & \;+\left(\frac{1}{\tau_{v}}A_{v}+\frac{1}{\tau }A_{v}^{-1}A_{vq}A_{qv}+A_{v}^{-1}A_{vx}A_{xv}\right)Y_{2}\\ \; & \;+A_{v}^{-1}A_{vx}\left(B_{x}-A_{xq}A_{f}^{-1}B_{q}\right)V+A_{v}^{-1}B_{v}\dot{V} \end{array} \end{array}$$
where $F_{s}:=A_{s}-A_{xq}A_{f}^{-1}A_{qx}$, $F_{q}:=A_{f}^{-1}A_{qx}\left(A_{s}-A_{xq}A_{f}^{-1}A_{qx}\right)$, $F_{v}:=A_{v}^{-1}A_{vx}\left(A_{s}-A_{xq}A_{f}^{-1}A_{qx}\right)$, and
$$\begin{array}{cc} \; & A_{s}:=\left[\begin{array}{cccc} A_{c} & -E_{c}C & 0_{h\times n-\nu } & 0_{h\times \bar{n}-\nu }\\ gBC_{c} & A+B(F_{2}-gH_{c}C) & BF_{1} & 0_{\nu \times \bar{n}-\nu }\\ 0_{n-\nu \times h} & GC & S & 0_{n-\nu \times \bar{n}-\nu }\\ 0_{\bar{n}-\nu \times h} & \bar{G}C & 0_{\bar{n}-\nu \times n-\nu } & \bar{S} \end{array}\right],\\ \; & A_{xq}:=\left[\begin{array}{cc} 0_{h\times l} & 0_{h\times l}\\ -gBC_{\xi } & gBC_{\xi }\\ 0_{n\times l} & 0_{n\times l}\\ 0_{\bar{n}\times l} & 0_{\bar{n}\times l} \end{array}\right],\;B_{x}:=\left[\begin{array}{cc} B_{c} & 0_{h\times 1}\\ gBD_{c} & gB\\ 0_{n\times 1} & 0_{n\times 1}\\ 0_{\bar{n}\times 1} & 0_{\bar{n}\times 1} \end{array}\right]. \end{array}$$

Note that the X-dynamics without the term involving $Y_{1}$ and $Y_{2}$ in (Equation 28) is the quasi-steady-state model (Equation 23). Then, with $\tilde{X}=X-X_{N}$, we have $\dot{\tilde{X}}=F_{s}\tilde{X}+A_{xq}Y_{1}+A_{xv}Y_{2}$, while the $Y_{1}$ and $Y_{2}$-dynamics of (Equation 28) are rewritten as
(29)$$\begin{array}{c} \begin{array}{cc} {\dot{Y}}_{1} & =F_{q}\tilde{X}+\left(\frac{1}{\tau }A_{f}+A_{f}^{-1}A_{qx}A_{xq}\right)Y_{1}\\ \; & \;+\left(\frac{1}{\tau }A_{qv}+A_{f}^{-1}A_{qx}A_{xv}\right)Y_{2}+B_{r}\theta \\ {\dot{Y}}_{2} & =F_{v}X+\left(\frac{1}{\tau }A_{v}^{-1}A_{vq}A_{f}+A_{f}^{-1}A_{qx}A_{xq}\right)Y_{1}\\ \; & \;+\left(\frac{1}{\tau_{v}}A_{v}+\frac{1}{\tau }A_{v}^{-1}A_{vq}A_{qv}+A_{v}^{-1}A_{vx}A_{xv}\right)Y_{2}+{\bar{B}}_{r}\theta \end{array} \end{array}$$
where $\theta :={\left[X_{N}^{T},V^{T},{\dot{V}}^{T}\right]}^{T}$,
$$\begin{array}{cc} \; & B_{r}:=\left[\begin{array}{ccc} F_{q} & A_{f}^{-1}A_{qx}\left(B_{x}-A_{xq}A_{f}^{-1}B_{q}\right) & A_{f}^{-1}B_{q} \end{array}\right],\\ \; & {\bar{B}}_{r}:=\left[\begin{array}{ccc} F_{v} & A_{v}^{-1}A_{vx}\left(B_{x}-A_{xq}A_{f}^{-1}B_{q}\right) & A_{v}^{-1}B_{v} \end{array}\right]. \end{array}$$

By Assumption 2 and 3, the matrix $F_{s}$ and $A_{v}$ are Hurwitz. If the matrix $A_{f}$ is Hurwitz, then there exist positive definite matrices $P_{f}$, $P_{s}$, and $P_{v}$, such that $P_{s}F_{s}+F_{s}^{T}P_{s}=-2I$, $P_{f}A_{f}+A_{f}^{T}P_{f}=-2I$, and $P_{v}A_{v}+A_{v}^{T}P_{v}=-2I$. Let $V_{2}\left(\tilde{X},Y_{1},Y_{2}\right)=\frac{1}{2}{\tilde{X}}^{T}P_{s}\tilde{X}+\frac{1}{2}Y_{1}^{T}P_{f}Y_{1}+\frac{1}{2}\delta Y_{2}^{T}P_{v}Y_{2}$, where a positive constant δ will be chosen later. Then, we obtain
$$\begin{array}{c} \begin{array}{cc} {\dot{V}}_{2} & \leq -∥\tilde{X}{∥}^{2}-\frac{1}{\tau }∥Y_{1}{∥}^{2}-\frac{\delta }{\tau_{v}}∥Y_{2}{∥}^{2}+\gamma_{1}∥\tilde{X}∥∥Y_{1}∥+\left(\gamma_{2}+\delta \gamma_{7}\right)∥\tilde{X}∥∥Y_{2}∥\\ \; & \;+\left(\gamma_{4}\frac{1}{\tau }+\gamma_{5}+\delta \frac{1}{\tau }\gamma_{8}+\delta \gamma_{9}\right)∥Y_{1}∥∥Y_{2}∥+\gamma_{3}{∥Y_{1}∥}^{2}\\ \; & \;+\left(\delta \frac{1}{\tau }\gamma_{10}+\delta \gamma_{11}\right)∥Y_{2}{∥}^{2}+\gamma_{6}∥Y_{1}∥+\delta \gamma_{12}∥Y_{2}∥ \end{array} \end{array}$$
where
$$\begin{array}{c} \begin{array}{cc} \; & \gamma_{1}=∥P_{s}A_{xq}+F_{q}^{T}P_{f}∥,\gamma_{2}=∥P_{s}A_{xv}∥,\gamma_{3}=∥P_{f}A_{f}^{-1}A_{qx}A_{xq}∥,\gamma_{4}=∥P_{f}A_{qv}∥,\\ \; & \gamma_{5}=∥P_{f}A_{f}^{-1}A_{qx}A_{xv}∥,\;\gamma_{6}=∥P_{f}B_{r}∥\max_{0\leq t\leq \infty }∥\theta \left(t\right)∥,\;\gamma_{7}=∥F_{v}^{T}P_{v}∥,\\ \; & \gamma_{8}=∥P_{v}A_{v}^{-1}A_{vq}A_{f}∥,\;\gamma_{9}=∥P_{v}A_{f}^{-1}A_{qx}A_{xq}∥,\;\gamma_{10}=∥P_{v}A_{v}^{-1}A_{vq}A_{qv}∥,\\ \; & \gamma_{11}=∥P_{v}A_{v}^{-1}A_{vx}A_{xv}∥,\;\gamma_{12}=∥P_{v}{\bar{B}}_{r}∥\max_{0\leq t\leq \infty }∥\theta \left(t\right)∥. \end{array} \end{array}$$

We choose δ such that $32\gamma_{2}^{2}{\bar{\tau }}_{v}\leq \delta \leq 1/\left(32\gamma_{7}^{2}{\bar{\tau }}_{v}\right)$ and assume ${\bar{\tau }}_{v}\leq 1/\left(16\gamma_{11}\right)$. It is possible because we already assume that the time constant of unmodeled dynamics, $\tau_{v}$, is sufficiently small. By Assumptions 1–3, values of $\gamma_{1}-\gamma_{12}$ also can be obtained. If we select τ that satisfies $\tau^{†}<\tau <\bar{\tau }$ where
$$\begin{array}{cc} \; & \tau^{†}:=\max\left\{16\gamma_{10}{\bar{\tau }}_{v},64\gamma_{4}^{2}\frac{1}{\delta }{\bar{\tau }}_{v},64\gamma_{8}^{2}\delta {\bar{\tau }}_{v}\right\},\\ \; & \bar{\tau }:=\min\left\{\frac{1}{8(2\gamma_{1}^{2}+\gamma_{3})},\frac{\delta }{64\gamma_{5}^{2}{\bar{\tau }}_{v}},\frac{1}{64\delta \gamma_{9}^{2}{\bar{\tau }}_{v}}\right\}, \end{array}$$
then it can be shown that ${\dot{V}}_{2}<0$ when $∥Y_{1}∥>2\gamma_{6}\tau$ and $∥Y_{2}∥>2\gamma_{12}\tau_{v}$. Define ${\bar{V}}_{2}:=\frac{1}{2}∥\tilde{X}{∥}^{2}+\frac{1}{\tau }∥Y_{1}{∥}^{2}+\frac{\delta }{\tau_{v}}{∥Y_{2}∥}^{2}$. Then, it holds that
$${\dot{V}}_{2}\leq -\frac{1}{2}{\bar{V}}_{2}+\frac{\sqrt{2}}{4}\bar{\gamma }\left(\tau ,\tau_{v}\right)\sqrt{{\bar{V}}_{2}}$$
where $\bar{\gamma }\left(\tau ,\tau_{v}\right):=4\max\left\{\gamma_{6},\gamma_{12}\right\}\cdot \max\left\{\sqrt{\tau },\sqrt{\tau_{v}/\delta }\right\}$. If $∥{\bar{V}}_{2}∥>\frac{1}{2}{\bar{\gamma }}^{2}\left(\tau ,\tau_{v}\right)$, then ${\dot{V}}_{2}<0$. And, a positive constant $\mu_{2}$ is given by
$$\mu_{2}:=\max_{{\bar{V}}_{2}=\frac{1}{2}{\bar{\gamma }}^{2}\left(\tau ,\tau_{v}\right)}V_{2}\left(\tilde{X},Y_{1},Y_{2}\right)$$

Now, we define the set $\Omega_{v}:=\left\{\left[\tilde{X};Y_{1};Y_{2}\right]|V_{2}\left(\tilde{X},Y_{1},Y_{2}\right)\leq \mu_{2}\right\}$. It is obvious that the state $\left[\tilde{X};Y_{1};Y_{2}\right]$ converges to the set $\Omega_{v}$ as t→∞. Also,
$$V_{2}=\frac{1}{2}{\tilde{X}}^{T}P_{s}\tilde{X}+\frac{1}{2}Y_{1}^{T}P_{f}Y_{1}+\frac{\delta }{2}Y_{2}^{T}P_{v}Y_{2}\leq \rho_{2}{\bar{V}}_{2}$$
where $\rho_{2}:=\max\left\{\lambda_{\max}\left(P_{s}\right),\frac{1}{2}\lambda_{\max}\left(P_{f}\right)\tau ,\frac{1}{2}\lambda_{\max}\left(P_{v}\right)\tau_{v}\right\}$. For all $\left[\tilde{X},Y_{1},Y_{2}\right]\in \Omega_{\alpha }$, $∥\tilde{X}{∥}^{2}\leq \mu_{2}\leq \frac{1}{2}\rho_{2}{\bar{\gamma }}^{2}\left(\tau ,\tau_{v}\right)$, and thus
$$∥\tilde{X}∥\leq \sqrt{\frac{\rho_{2}}{2}}\bar{\gamma }\left(\tau ,\tau_{v}\right).$$

Since we assume that $\tau_{v}$ is relatively small positive constant, $\rho_{2}=\lambda_{\max}\left(P_{s}\right)/2$ as τ is reduced and the bound of $∥X-X_{N}∥$ is proportional to the $\bar{\gamma }\left(\tau ,\tau_{v}\right)$, it means that $V_{2}$ tends to be small depending on τ, so that the error $X\left(t\right)-X_{N}\left(t\right)$ becomes small, by taking τ appropriately. By summarizing the above results, we provide the following theorem.

**Theorem** **2.**
*Under Assumptions 1–3, for a sufficiently small ${\bar{\tau }}_{v}$, there exist positive constants $\tau^{†}=\max\left\{16\gamma_{10}{\bar{\tau }}_{v},64\gamma_{4}^{2}\frac{1}{\delta }{\bar{\tau }}_{v},64\gamma_{8}^{2}\delta {\bar{\tau }}_{v}\right\}$ and $\bar{\tau }=\min\left\{\frac{1}{8(2\gamma_{1}^{2}+\gamma_{3})},\frac{\delta }{64\gamma_{5}^{2}{\bar{\tau }}_{v}},\frac{1}{64\delta \gamma_{9}^{2}{\bar{\tau }}_{v}}\right\}$ such that, for all $\tau^{†}<\tau <\bar{\tau }$, the overall closed-loop system *(Equation 17)* and *(Equation 18)* is exponential stable when $y_{r}=0$ and d=0. Furthermore, the part of solution of *(Equation 17)* and *(Equation 18)* denoted by $\left[c\left(t\right);x\left(t\right);z\left(t\right);\bar{z}\left(t\right)\right]$ satisfies that*

$$\begin{array}{cc} \underset{t\to \infty }{lim sup} & ∥\left[c\left(t\right);x\left(t\right);\bar{z}\left(t\right)\right]\;-\;\left[c_{N}\left(t\right);x_{N}\left(t\right);{\bar{z}}_{N}\left(t\right)\right]∥\;\leq \;\Gamma_{2}\bar{\gamma }\left(\tau_{q},\tau_{v}\right) \end{array}$$

*where $\Gamma_{2}:=\sqrt{\rho_{2}/2}$ and $\left[c_{N}\left(t\right);x_{N}\left(t\right);{\bar{z}}_{N}\left(t\right)\right]$ is the solution of the nominal closed-loop system *(Equation 4)* and *(Equation 5)*.*


## 5. Simulation Results

In this section, we provide simulation results for mass-damper systems to validate the effectiveness of the proposed performance and stability analysis. The actual plant P and its nominal model $P_{n}$ are represented as
(30)$$\begin{array}{c} \begin{array}{cc} \dot{x} & =\left[\begin{array}{cc} 0 & 1\\ 0 & -2 \end{array}\right]x+\left[\begin{array}{c} 0\\ 4 \end{array}\right]\left(u+d\right),\\ y & =\left[\begin{array}{cc} 1 & 0 \end{array}\right]x \end{array} \end{array}$$
and
(31)$$\begin{array}{c} \begin{array}{cc} {\dot{x}}_{n} & =\left[\begin{array}{cc} 0 & 1\\ 0 & -3 \end{array}\right]x_{n}+\left[\begin{array}{c} 0\\ 1 \end{array}\right]u_{r},\\ y_{n} & =\left[\begin{array}{cc} 1 & 0 \end{array}\right]x_{n}. \end{array} \end{array}$$

We assume that the unmodeled dynamics is given by
(32)$$\begin{array}{c} \begin{array}{cc} \tau_{v}\dot{v} & =\left[\begin{array}{cc} 0 & 1\\ -1 & -2 \end{array}\right]x_{n}+\left[\begin{array}{c} 0\\ 1 \end{array}\right]u_{v},\\ u & =\left[\begin{array}{cc} 1 & 0 \end{array}\right]v. \end{array} \end{array}$$

The outer-loop controller C is designed as
(33)$$\begin{array}{c} u_{r}=3\left(y_{r}-y\right), \end{array}$$
and two Q-filters, $Q_{q}$ and $Q_{p}$, are presented as
(34)$$\begin{array}{c} \begin{array}{cc} \tau_{q}\dot{q} & =\left[\begin{array}{cc} 0 & 1\\ -1 & -2 \end{array}\right]q+\left[\begin{array}{c} 0\\ 1 \end{array}\right]y,\\ \bar{w} & =\left[\begin{array}{cc} 1 & 0 \end{array}\right]q, \end{array} \end{array}$$
(35)$$\begin{array}{c} \begin{array}{cc} \tau_{q}\dot{p} & =\left[\begin{array}{cc} 0 & 1\\ -1 & -2 \end{array}\right]p+\left[\begin{array}{c} 0\\ 1 \end{array}\right]u_{v},\\ \hat{u} & =\left[\begin{array}{cc} 1 & 0 \end{array}\right]p. \end{array} \end{array}$$

Figure 2 shows simulation results of the disturbance observer-based control system under the unmodeled dynamics with respect to the time constant of Q-filter $\tau_{q}$. The cyan dotted and green solid lines represent the outputs of the nominal closed-loop system composed of (Equation 31) and (Equation 33) and the real closed-loop system (Equation 30), (Equation 32), and (Equation 33), respectively. The blue dashed, red dash-dotted, and black solid lines are the outputs of the disturbance observer-based control system with $\tau_{q}=0.1,0.2$, and 0.3, respectively. To clarify the explanation, the errors between the output of the nominal closed-loop system and the outputs of the disturbance observer-based control system with $\tau_{q}=0.1,0.2$, and 0.3, as provided in Figure 3. As can be seen in Figure 2, in the absence of the disturbance observer, the effect of plant uncertainties and disturbances degrades the control performance. However, as shown in Figure 2 and Figure 3, the disturbance observer completely rejects the effect of the constant disturbance in the steady state, and the disturbance rejection performance improves as $\tau_{q}$ decreases. Moreover, the disturbance observer recovers the nominal performance and the error asymptotically converges to a set whose size is proportional to the square root of $\tau_{q}$.

On the other hand, Figure 4 and Figure 5 present simulation results for smaller $\tau_{q}$. The blue dashed, red dash-dotted, and black solid lines are the outputs and errors of the disturbance observer-based control system with $\tau_{q}=0.05,0.06$, and 0.12, respectively. As shown in Figure 4 and Figure 5, the closed-loop system becomes unstable when $\tau_{q}$ is smaller than the stability bound. To figure out the effect of $\tau_{v}$ on the stability of the closed-loop system, the errors of the disturbance observer-based control system with respect to $\tau_{v}$ are shown in Figure 6. It can be observed that the larger $\tau_{v}$ makes the closed-loop system unstable.

## 6. Conclusions

In this paper, we investigated the robust stability and the nominal performance recovery of the disturbance observer-based control system with unmodeled dynamics using the singular perturbation theory. Furthermore, the explicit bound of the time constant of the Q-filter for robust stability of the closed-loop system was provided based on the Lyapunov analysis. The obtained results can be applied to the plant with unstable poles, as well as the Q-filter of arbitrary relative degree whose coefficients are not limited to the binomial one. Finally, we showed that the disturbance observer recovers a nominal performance, which is designed for the nominal model of the plant and the state error between the nominal and the actual closed-loop system asymptotically converges to a set whose size is proportional to the square root of the time constant of the Q-filter.

## Figures and Tables

**Figure 1 sensors-24-07850-f001:**
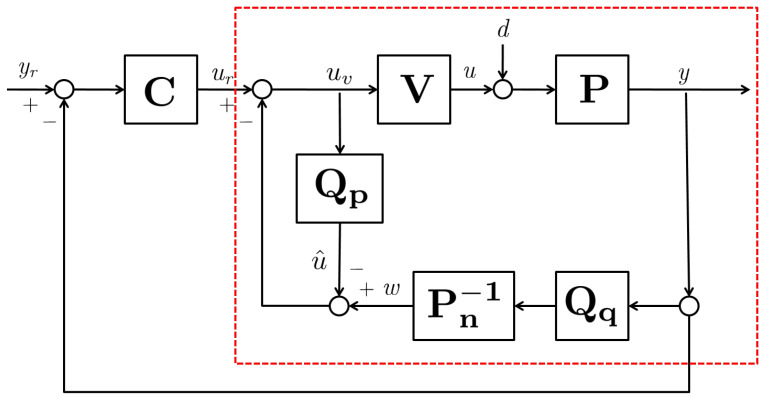
The configuration of the disturbance observer-based control system.

**Figure 2 sensors-24-07850-f002:**
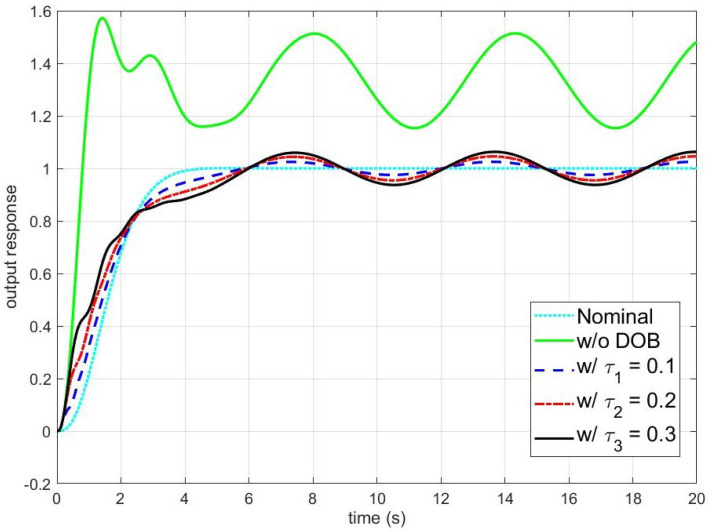
The output response of the disturbance observer-based control system under the unmodeled dynamics (Nominal: the output of the nominal closed-loop system; w/o DOB: the output of the real closed-loop system without the disturbance observer; w/$\tau_{1}$: the output with $\tau_{1}=0.1$; w/$\tau_{2}$: the output with $\tau_{2}=0.2$; and w/$\tau_{3}$: the output with $\tau_{3}=0.3$). The reference input $y_{r}\left(t\right)$, the disturbance d(t), and the time constant of the unmodeled dynamics $\tau_{v}$ are $y_{r}\left(t\right)=\tanh\left(t\right)$, d(t)=1+0.5sin(t), and $\tau_{v}=0.02$, respectively.

**Figure 3 sensors-24-07850-f003:**
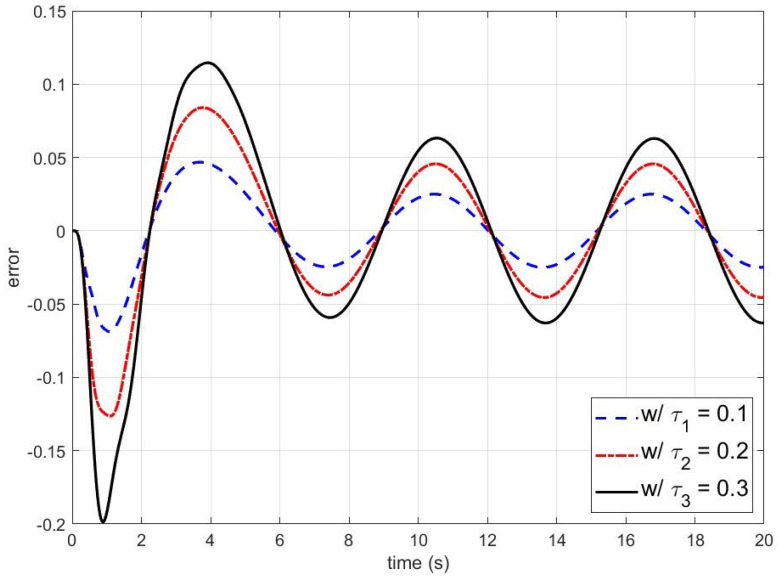
The errors between the output of the nominal closed-loop system and the output of the disturbance observer-based control system (w/$\tau_{1}$: the error with $\tau_{1}=0.1$; w/$\tau_{2}$: the error with $\tau_{2}=0.2$; and w/$\tau_{3}$: the error with $\tau_{3}=0.3$). The reference input $y_{r}\left(t\right)$, the disturbance d(t), and the time constant of the unmodeled dynamics $\tau_{v}$ are $y_{r}\left(t\right)=\tanh\left(t\right)$, d(t)=1+0.5sin(t), and $\tau_{v}=0.02$, respectively.

**Figure 4 sensors-24-07850-f004:**
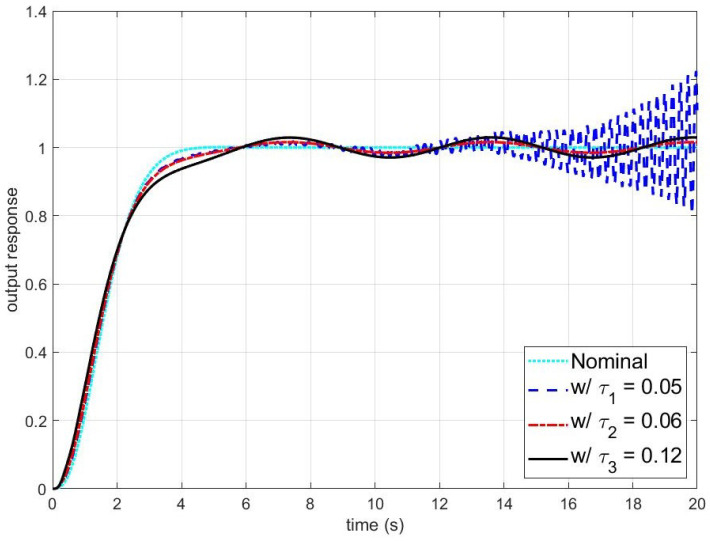
The output response of the disturbance observer-based control system under the unmodeled dynamics (Nominal: the output of the nominal closed-loop system; w/$\tau_{1}$: the output with $\tau_{1}=0.05$; w/$\tau_{2}$: the output with $\tau_{2}=0.06$; and w/$\tau_{3}$: the output with $\tau_{3}=0.12$). The reference input $y_{r}\left(t\right)$, the disturbance d(t), and the time constant of the unmodeled dynamics $\tau_{v}$ are $y_{r}\left(t\right)=\tanh\left(t\right)$, d(t)=0.5sin(t), and $\tau_{v}=0.02$, respectively.

**Figure 5 sensors-24-07850-f005:**
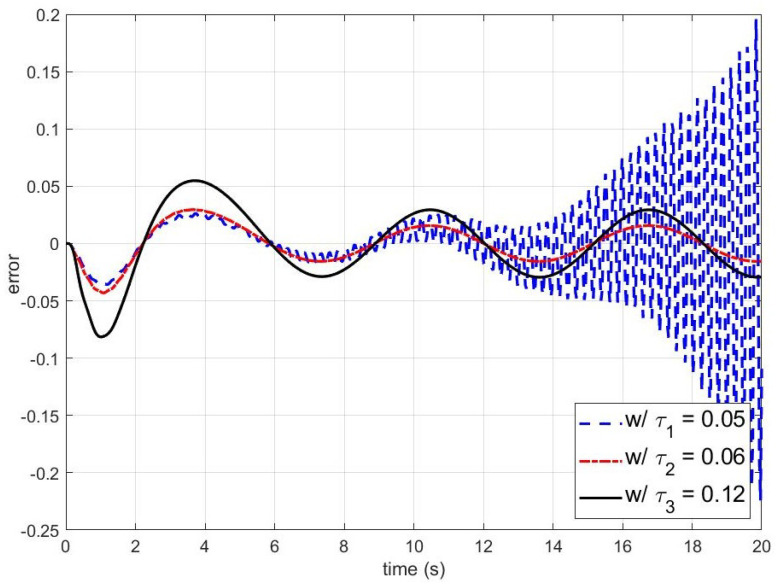
The errors between the output of the nominal closed-loop system and the output of the disturbance observer-based control system (w/$\tau_{1}$: the error with $\tau_{1}=0.05$; w/$\tau_{2}$: the error with $\tau_{2}=0.06$; and w/$\tau_{3}$: the error with $\tau_{3}=0.12$). The reference input $y_{r}\left(t\right)$, the disturbance d(t), and the time constant of the unmodeled dynamics $\tau_{v}$ are $y_{r}\left(t\right)=\tanh\left(t\right)$, d(t)=0.5sin(t), and $\tau_{v}=0.02$, respectively.

**Figure 6 sensors-24-07850-f006:**
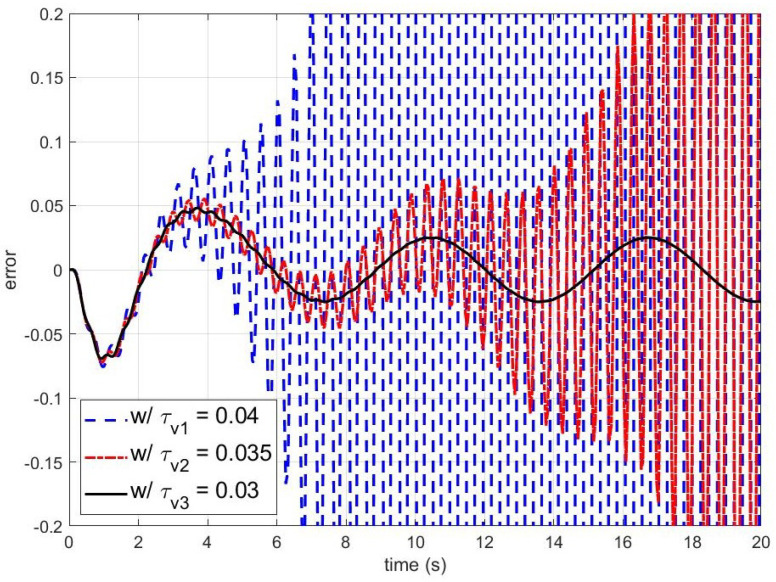
The errors between the output of the nominal closed-loop system and the output of the disturbance observer-based control system with different time constants of the unmodeled dynamics (w/$\tau_{v1}$: the error with $\tau_{v1}=0.04$; w/$\tau_{v2}$: the error with $\tau_{v2}=0.035$; and w/$\tau_{v3}$: the error with $\tau_{v3}=0.03$). The reference input $y_{r}\left(t\right)$, the disturbance d(t), and the time constant of the Q-filter $\tau_{q}$ are $y_{r}\left(t\right)=\tanh\left(t\right)$, d(t)=0.5sin(t), and $\tau_{q}=0.01$, respectively.

## Data Availability

Data are contained within the article.

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
