# Peer review of "Performance Recovery and Stability Analysis of Disturbance Observer Under Unmodeled Dynamics"

_sensors, 2024, doi:10.3390/s24237850_

Round 1
Reviewer 1 Report
Comments and Suggestions for Authors
The research question sounds good. However, it would be reasonable to see the experiment validation of the simulation results. A case with only one value of a time constant for unmodelled dynamics is shown, but how about other values? What is the upper bound for the time constant? More simulation and experimental results are required. Thanks!
Author Response
Comments 1: The research question sounds good.
Response 1: Thank the reviewer for the comments.
Comments 2: However, it would be reasonable to see the experiment validation of the simulation results.
Response 2: Thank the reviewer for the constructive comments. We are trying to implement the motor control system for the experimental validation. Unfortunately, we have not finished the experiment setup yet, so we will continue the hardware implementation in future work.
Comments 3: A case with only one value of a time constant for unmodelled dynamics is shown, but how about other values? What is the upper bound for the time constant? More simulation and experimental results are required. Thanks!
Response 3: The simulation section has been revised to validate the effectiveness of the proposed performance and stability analysis. As can be seen in Theorem 2, the upper bound of the time constant of the unmodeled dynamics is determined by the time constant of the Q-filter and plant uncertainties. However, the presented explicit bound is derived from the Lyapunov analysis, which is conservative. Therefore, Fig. 6 has been included in the simulation section to figure out the effect of the time constant of the unmodeled dynamics on the stability of the closed-loop system.
Reviewer 2 Report
Comments and Suggestions for Authors
This paper reveals the robustness and disturbance rejection performance of the disturbance observer based on the singular perturbation theory and proposes its design guideline for robust stability in the presence of unmodeled dynamics. In addition, this paper presents that the disturbance observer recovers a nominal performance designed for a nominal model of the plant.
1) The designed observer is normally compared with the existing ones. What is the superiority?
2) What kinds of disturbance can be estimated by the designed observer?
3) The main contribution needs to be discussed clearly since it is confusing.
4) More comparisons need to be conducted with the existing results from simulation or theory.
5) What is the practical application of the proposed method?
6) The background of the observer is insufficient, some references such as Zhu Automatica 147 (2023): 110744 and Zhang IEEE/CAA Journal of Automatica Sinica 11.3 (2024): 661-672, and Xu IEEE Transactions on Industrial Electronics (2024).
Author Response
Overall Comments: This paper reveals the robustness and disturbance rejection performance of the disturbance observer based on the singular perturbation theory and proposes its design guideline for robust stability in the presence of unmodeled dynamics. In addition, this paper presents that the disturbance observer recovers a nominal performance designed for a nominal model of the plant.
Response: Thank the reviewer for spending time and effort on the evaluation of our paper.
Comments 1: The designed observer is normally compared with the existing ones. What is the superiority?
Response 1: In order to clarify the superiority of the disturbance observer discussed in this paper, we have revised the second paragraph of Introduction. In specific, among various disturbance compensation techniques, a disturbance observer has been widely used in the industrial field because of its versatility. First, it has a simple structure composed of the inverse dynamics of the nominal model of the actual plant and two low-pass filters (known as Q-filter). Secondly, it is convenient to use since it is an add-on type inner-loop controller. In other words, if it is added to the inner loop, then the existing (pre-designed outer-loop) controller is enabled without taking into account effects from disturbances and plant uncertainties. Thirdly, it effectively compensates the plant uncertainties and rejects external disturbances without knowledge of the disturbance model. Finally, the control designer could easily improve the disturbance rejection performance merely by decreasing the time constant of the Q-filter.
Comments 2: What kinds of disturbance can be estimated by the designed observer?
Response 2: The presented disturbance observer the effect of the unmodeled disturbance approximately, and its disturbance rejection performance improves as the time constant of the Q-filter decreases. However, by embedding the disturbance model into the disturbance observer structure, the effect of modeled disturbances can be completely removed. The Q-filter design methods have been proposed to cope with polynomial-in-time disturbances, such as the step, ramp, parabolic signals, and so on, as well as sinusoidal-type disturbances with unknown amplitudes and phases. Thus, the disturbance observer can reject not only unmodeled but bounded disturbances approximately but also modeled disturbances asymptotically. This explanation has been added to the second paragraph of Introduction.
Comments 3: The main contribution needs to be discussed clearly since it is confusing.
Response 3: The fifth paragraph of Introduction has been modified to clarify the contribution of this paper. Specifically, the contribution of this paper is as follows. First, taking into account the unmodeled dynamics in the closed-loop system, the performance and stability analysis of the disturbance observer-based control system is presented using the singular perturbation theory. Secondly, based on the Lyapunov analysis, robust stability conditions under plant uncertainties are derived. Especially, the explicit bound of the time constant of the Q-filter, which guarantees robust stability of the closed-loop system, is provided. Finally, it is shown that the disturbance observer recovers a nominal performance, which is designed for the nominal model of the plant and the state error between the nominal and the actual closed-loop system asymptotically converges to a set whose size is proportional to the square root of the time constant of the Q-filter.
Comments 4: More comparisons need to be conducted with the existing results from simulation or theory.
Response 4: Thank you for the constructive comments. The first paragraph of Introduction has been revised to provide a comparison with the existing results and the presented disturbance observer. The disturbance estimation and compensation problem is one of the important issues in the control community and has been discussed in many applications such as the motor control systems [1-6], robot manipulator systems [7-9], wheeled mobile robot systems [10], and positioning stage systems [11]. To improve the disturbance rejection performance and robustness, various approaches have been investigated. For instance, an unknown input observer estimates the unknown external signal based on the internal model principle, assuming that the unknown input or disturbance is generated by some exosystem [12, 13]. By defining the lumped disturbance as a new state variable, an extended state observer simultaneously estimates the augmented state [14, 15]. Further, by incorporating the extended state observer into the control structure, an active disturbance rejection control technique has been presented, and it simplifies the uncertain system with lumped disturbances as a disturbance-free nominal linear system of the form of the integrator chain [16, 17]. Instead of canceling the nonlinear term in the nonlinear system, a nonlinear disturbance observer has been proposed to include the known or partially known nonlinear part in the observer design [18, 19]. Several survey papers have been presented to figure out the pros and cons of each technique [20-22].
Moreover, in the simulation section, Fig. 2 has been added to the revised version of the paper to provide a comparison between the output response of the closed-loop system with and without the disturbance observer.
Comments 5: What is the practical application of the proposed method?
Response 5: The presented performance and stability analysis is applicable to the control system, which has the actuator and sensor dynamics. In this paper, the simulations have been performed for the mechanical system composed of the mass and damping term, taking into account the unmodled dynamics.
Comments 6: The background of the observer is insufficient, some references such as Zhu Automatica 147 (2023): 110744 and Zhang IEEE/CAA Journal of Automatica Sinica 11.3 (2024): 661-672, and Xu IEEE Transactions on Industrial Electronics (2024).
Response 6: Thank the reviewer for the constructive comment. In accordance with the reviewer's comment, to sufficiently explain the state-of-the-art research on the disturbance observer, we have added the following five papers in the reference.
[3] Li, Q.; Li, S.; Xu, Y.; Yu, G.; Gao, Y.; Zou, J. Active inertia extended resonance ratio control for permanent magnetic coupling transmission system. IEEE Trans. Ind. Electron. 2024, 71, 11989-12000.
[4] Tian, M.; Wang, B.; Yu, Y.; Dong, Q.; Xu, D. Adaptive active disturbance rejection control for uncertain current ripples suppression of PMSM drives. IEEE Trans. Ind. Electron. 2024, 71, 2320-2331.
[5] Li, Q.; Li, H.; Gao, J.; Xu, Y.; Rodriguez, J.; Kennel, Ralph. Nonlinear-disturbance-observer-based model-predictive control for servo press drive. IEEE Trans. Ind. Electron. 2024, 71, 8448-8458.
[8] Zhang, D.; Hu, J.; Cheng, J.; Wu, Z.; Yan, H. A novel disturbance observer based fixed-time sliding mode control for robotic manipulators with global fast convergence. IEEE/CAA J. Automatica Sinica 2024, 11, 661-672.
[13] Zhu, F.; Fu, Y.; Dinh, T. N. H. Asymptotic convergence unknown input observer design via interval observer. Automatica 2023, 147, 110744.
Reviewer 3 Report
Comments and Suggestions for Authors
edit the structure of the article introduction, methods, results, conclusion
Author Response
Comments 1: edit the structure of the article introduction, methods, results, conclusion
Response 1: Thank the reviewer for the comments. In order to clarify the contribution of the paper, the first, second, and fifth paragraphs of Introduction have been revised. Furthermore, the simulation section has also been revised to clearly explain the effectiveness of the proposed performance and stability analysis.
Round 2
Reviewer 1 Report
Comments and Suggestions for Authors
Thanks for the clarifications. I am satisfied with the results.